# Protected areas network is not adequate to protect a critically endangered East Africa Chelonian: Modelling distribution of pancake tortoise, *Malacochersus tornieri* under current and future climates

Abraham Eustace[1]*, Luíz Fernando Esser[2], Rudolf Mremi[3], Patrick K. Malonza[4], Reginald T. Mwaya[3]

**1** Tanzania Wildlife Management Authority, Morogoro, Tanzania, **2** Laboratório de Fitoecologia e Fitogeografia, Programa de Pós-Graduação em Botânica, Universidade Federal do Rio Grande do Sul, Porto Alegre, RS, Brazil, **3** College of African Wildlife Management, Mweka, Moshi, Tanzania, **4** Herpetology Section, National Museums of Kenya, Nairobi, Kenya

\* abrah15@gmail.com

**Data Availability Statement:** According to IUCN the pancake tortoise is critically endangered and threatened by international pet trade therefore the

## Abstract

While the international pet trade and habitat destruction have been extensively discussed as major threats to the survival of the pancake tortoise (*Malacochersus tornieri*), the impact of climate change on the species remains unknown. In this study, we used species distribution modelling to predict the current and future distribution of pancake tortoises in Zambezian and Somalian biogeographical regions. We used 224 pancake tortoise occurrences obtained from Tanzania, Kenya and Zambia to estimate suitable and stable areas for the pancake tortoise in all countries present in these regions. We also used a protected area network to assess how many of the suitable and stable areas are protected for the conservation of this critically endangered species. Our model predicted the expansion of climatically suitable habitats for pancake tortoises from four countries and a total area of 90,668.75 km² to ten countries in the future and an area of 343,459.60–401,179.70 km². The model also showed that a more significant area of climatically suitable habitat for the species lies outside of the wildlife protected areas. Based on our results, we can predict that pancake tortoises may not suffer from habitat constriction. However, the species will continue to be at risk from the international pet trade, as most of the identified suitable habitats remain outside of protected areas. We suggest that efforts to conserve the pancake tortoise should not only focus on protected areas but also areas that are unprotected, as these comprise a large proportion of the suitable and stable habitats available following predicted future climate change.

**Funding:** The author(s) received no specific funding for this work.

**Competing interests:** The authors have declared that no competing interests exist.

## Introduction

Over the past few decades, there has been growing interest in species distribution models (SDMs) as fundamental tools for the studies of ecology, biogeography, and biodiversity conservation [1–4]. These models are used to enhance understanding of the factors that alter species distribution, which is critical for adjusting and designing appropriate conservation strategies under current and future climatic scenarios [3, 5, 6]. Such adjustments are necessary because climate change poses a severe threat to the conservation of natural landscapes and species across the globe and is reported to be among the primary drivers of the current loss of global biodiversity [7–9]. Climate change has also been reported to accelerate shifts in range extension [7] and the shrinkage of some species [6, 10–12].

Tropical environments are widely recognized as biodiversity regions [13] with ideal climatic conditions for the survival of different species including reptiles [14]. However, reptiles are currently facing severe threats because of climatic changes [6, 15]. Because reptiles are sensitive to environmental change, it is undeniably that climate change affects reptile biodiversity directly by altering their distribution patterns [6, 7] and indirectly by threatening conservation areas, making them less habitable for reptiles [16]. For instance, Meng et al. [15] have reported that out of the 274 Tanzania reptile species they studied, 71% (194 reptile species) are vulnerable to climate change, suggesting that climate change affects reptilian diversity both directly or indirectly. In a different study, predictions about the environmental responses of reptiles to future climatic conditions made using SDMs showed that four endemic Moroccan reptilian species are highly vulnerable to extinction in Morocco if climatic disturbance prevails as predicted [6]. The same study concluded that reductions in species-rich areas is also likely in future climatic scenarios [6].

Like other reptiles, *Malacochersus tornieri* hereafter referred to as the pancake tortoise, is not immune to the effects of climate change. The pancake tortoise is a small, soft-shelled, dorsoventrally flattened chelonian with discontinuous distribution in the scattered rocky hills and kopjes of the savannas of south-eastern and northern Kenya and northern, eastern, and central Tanzania [17–20]. The presence of pancake tortoises has also been reported in northern Zambia [21]. The areas in which pancake tortoises can be found are typically semi-arid; these areas are classified as having a dry climate, corresponding to both Zambezian and Somalian biogeographic regions, according to Linder et al. [22]. The Zambezian biogeographical region is a wider biogeographical region, spreading across Africa from Namibia to Tanzania, while the Somalian biogeographic region is considered a refugium for arid-adapted plants and a centre of endemism for wide-range of animal taxa including reptiles [23, 24].

Although the international animal trade and habitat destruction have been cited as the major threats to the survival of the pancake tortoise [18, 19, 25, 26], the impact of climate change on the species remains largely unknown. Despite the fact that the IUCN has identified climate change as one of the threats to pancake tortoise populations [18, 19], to our understanding, there is no study that has assessed the impact of climate change on the future distribution pattern of pancake tortoises. Understanding these climatic patterns is one of the important steps in setting appropriate plans for re-introductions and translocations of the species which are important activities for conservation of species with threatened populations or restricted range. Furthermore, the IUCN's *Guidelines for Re-Introductions and Other Conservation Translocations* [27] has indicated climate-matching of recipient sites is important for understanding suitability of these areas for introduced/translocated species. Considering that the pancake tortoise is critically endangered [17–20] and listed in the *CITES Appendix II* [21], understanding current and future climatic habitats suitable for this species could be an essential step in charting out a realistic conservation plan for the species. Therefore, in this study,

we used species distribution modeling (SDM) to determine current and future climatic habitats suitable for the pancake tortoise. Identifying these climatic suitable habitats, might help to avoid uncertainties in selecting areas for translocation or introduction while providing a higher chance of success [27–29].

With time, the ongoing impacts of climate change are expected to inflict changes to suitable habitats for pancake tortoises both within and outside of protected areas [16, 30, 31]. While protected areas remain an essential approach for conserving and protecting biodiversity against human-mediated threats [15, 32], it could be challenging to protect the endangered species that inhabit areas outside of protected lands [33] such as Kenyan pancake tortoise [18]. For the development of specific and appropriate management and conservation plans for pancake tortoises, it is crucial to understand whether these protected areas will continue to be viable for protecting suitable habitats for the species in the event of climate change. Considering species' range varies under different climatic scenarios [3, 5, 34, 35] while the size of most protected areas tends to remain the same [36], more species may eventually be placed at risk of extinction, especially threatened species [37]. Therefore, in order to align protected areas with suitable habitat ranges [38] and enhance the conservation of threatened species in different climatic scenarios [36], SDMs can be used.

SDMs have been used to assess the impact of climate change on the distribution of different species (e.g. [30, 31, 36, 39]). These models use location data and environmental variables to predict the suitable distributional range of a species under climate change conditions [30, 36], which is essential when designing adequate species management programmes, as well as for endangered species conservation planning [40]. Although Bombi et al. [41] have used SDMs to model the distribution of all African tortoise species, including the pancake tortoise, their study did not predict the future distribution of the species. In this study, we used SDM to assess the distribution of pancake tortoises under current and future climatic conditions and to investigate how much of the climatically suitable habitat occurs within the Protected Areas Network in the Somali-Maasai and Zambezian biogeographical regions. Specifically, we assessed (i) the current and future climatically suitable areas for pancake tortoises, (ii) the occurrence of climatically persistent areas over time (henceforward, stable areas) and (iii) whether the protected areas will be viable for the conservation of the species. This study may inform species management approaches [42], including identifying suitable areas for translocation [29, 43, 44] and the establishment of nature reserves where species can be protected with minimal human intervention [10].

## Methodology

### Study area

We predicted current and future climatically suitable habitats for pancake tortoises within the two major biogeographical regions of Africa in which the animal occurs naturally (Fig 1B). These regions are the Somali-Masai Regional Centre of Endemism (Somali-Masai RCE) and the Zambezian Regional Centre of Endemism (Zambezian RCE), both of which fall within the semi-arid climatic belt of eastern-south Africa [45, 46]. The Somali-Masai RCE covers approximately 1.87 million km$^2$ of arid savannah, extending from north-eastern Somalia to the north-eastern province of Kenya and reaching south through Tanzania into the valley of the Great Ruaha; it ends north of Lake Malawi [22, 46, 47]. The Somali-Masai RCE harbours approximately 4,500 plant species, of which 31.00% are endemic in the region [45, 46]. The dominant vegetation in this region is *Acacia* spp. The Zambezian RCE (3.77 million km$^2$) extends in the northeast from the Somali-Masai RCE, and its distribution coincides with the Guinea savannas and woodlands and the Karoo-Namib RCE in the southwest [37, 38]. It

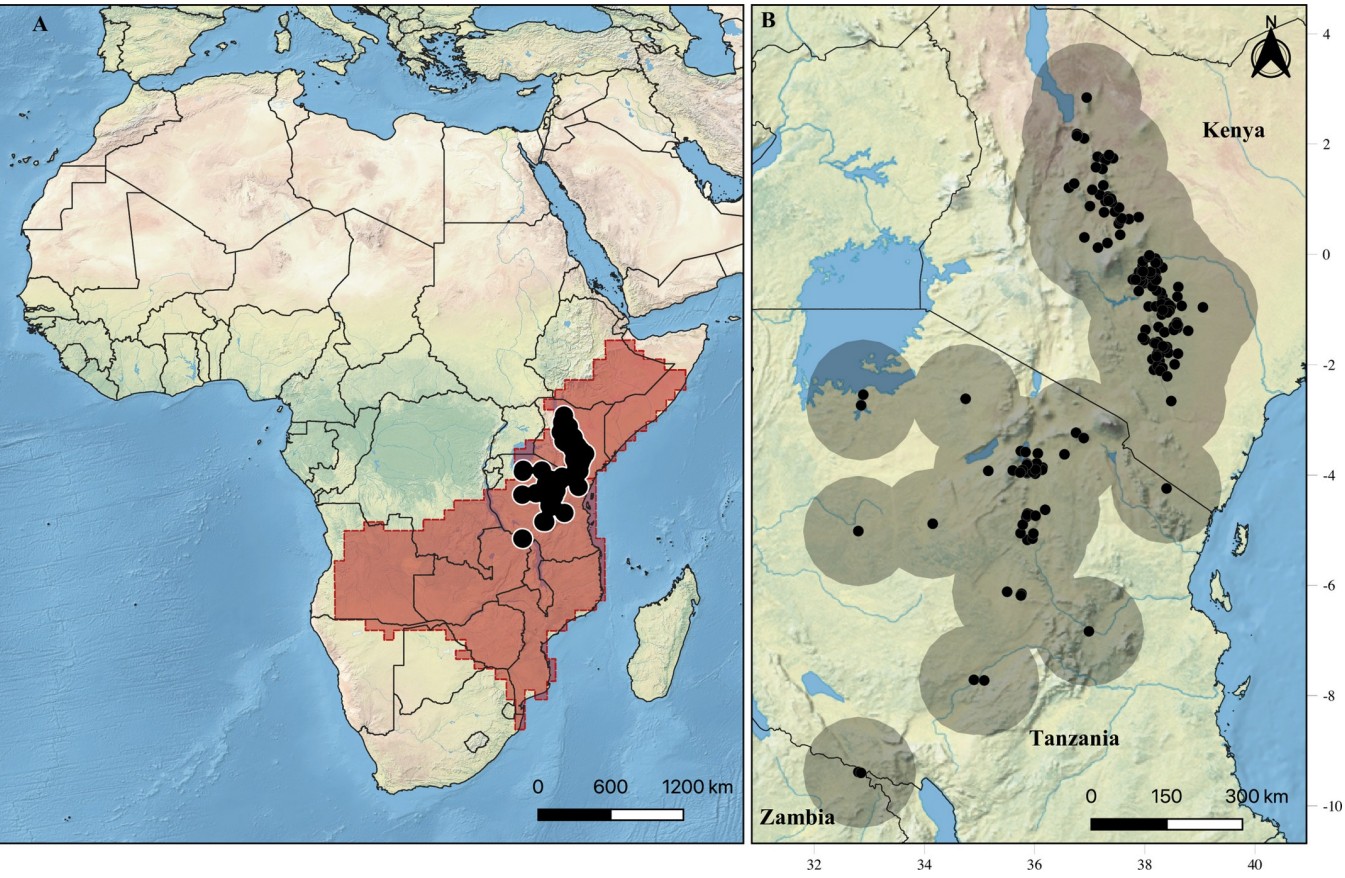

**Fig 1.** Current natural occurrence of pancake tortoise (*Malacochersus tornieri*), based on the data obtained in this study, with a 1-degree wide buffer around each presence record, at (A) continental scale with the Zambezian and Somalia biogeographical region (overlaid red polygon) and (B) regional scale. Background image was accessed from Natural Earth (public domain): http://www.naturalearthdata.com/.

covers the whole of south-central Africa, from the Atlantic seaboard of Angola to the entirety of Mozambique, Tanzania and the uplands of Kenya and Ethiopia [22, 47]. In terms of plant richness, it is more diverse than the Somali-Masai RCE, hosting about 8,500 plant species, out of which 54.00% are endemic in the region [45, 46].

## Study species

The critically endangered pancake tortoise, *Malacochersus* [20], is a monotypic genus endemic to East Africa [18, 48]. In East Africa, the pancake tortoise is restricted to Somali-Maasai and Zambezian vegetations [20, 26, 49–51]. In Tanzania, the species distribution is discontinuously scattered from the south-eastern shores of Lake Victoria to the Maasai Steppe and southward to Ruaha National Park [18, 20, 26]. In Kenya, the distribution of pancake tortoises is disconnected from the northern to southern areas, lying between central to south-eastern regions of the country [18, 20]. In Zambia, the species has been recorded only in the northern Nakonde District that borders Tanzania [20, 21]. The preferred micro-habitats for pancake tortoises are kopjes, rock outcrops and rocky hillsides [17, 18, 26] with an annual rainfall of 250–500 mm [50] and an elevational range of 400–1,600 m above sea level [52]. From the past two generations to the next generation, the observed and expected population of pancake tortoises is expected to decline by 80.00%, with overexploitation and habitat destruction being the

primary drivers [20]. Currently, the IUCN identifies biological resource use (intentional use) and agriculture (small-holder farming), as well as climate change (severe drought), as among the major threats to the habitat and population of pancake tortoises [20].

## Pancake tortoise occurrence data

We obtained occurrence data from the field, online databases and previous studies. We collected pancake tortoise location data from eight sites in Tarangire National Park, three sites in the Babati district of the Manyara region, five sites in the Kondoa districts and two sites in Chemba district, both from the Dodoma region in central Tanzania. The permit for conducting field work was granted by Commission for Science and Technology (COSTECH) and Tanzania Wildlife Research Institute (TAWIRI) while free access to protected areas was granted by Tanzania National Parks (TANAPA).

We also downloaded pancake tortoise locations from the GBIF (https://www.gbif.org/) and VertNet (http://vertnet.org/) by using rgbif [53] and rvertnet [54] R packages respectively. Both databases were accessed on 5 January 2020, and we downloaded all *Malacochersus tornieri* locations identified in Tanzania and Kenya. We did not find any pancake tortoise occurrences in Zambia in the two databases. From the online databases, we excluded data with absent or incomplete coordinates and duplicate locations as well as non-natural locations, such as tortoise collection points, captive breeding sites and pet-animal release sites. Additionally, we searched for pancake tortoise locality records in the EMYSystem Global Turtle Database [55] and then used Elevation Map (https://elevationmap.net/) to obtain location coordinates.

From previous studies, we extracted the names of the places where pancake tortoises were recorded/observed. For Tanzania, we used sites mentioned by Klemens and Moll [26] as well as point locations collected by Zacarias [56], while for Zambia we used point locations mentioned by Chansa and Wagner [21]. In Kenya, we obtained pancake tortoise sites from Malonza [18] and Kyalo [52]. After obtaining the site names, we used Google Maps (https://www.google.co.tz/maps/), Elevation Map (https://elevationmap.net/) and Mindat (https://www.mindat.org/) to obtain coordinates for each site. If the site was not available online, we contacted individuals currently or previously working in the area in order to obtain coordinates. From all sources, we obtained data for a total of 224 occurrences, with most occurrence points falling within the current IUCN pancake tortoise distribution range (Fig 1).

## Bioclimatic variables

Nineteen bioclimatic variables (BIO 1–19) were obtained from the CHELSA database [57] with 30 arc-seconds resolution. The modelling domain comprised Zambezian and Somalian biogeographical regions [22]. These regions were selected because they represent the areas where pancake tortoises exist naturally [20, 21, 26, 49–51]. We obtained variables for the two intermediate Representative Concentration Pathways (RCPs), RCP 4.5 [58] and RCP 6.0 [59], for the years 2050 (mean climate between 2041 and 2060) and 2070 (mean climate between 2061 and 2080). These mid-impact RCPs are the most desirable for future conservation planning, since they present a more realistic path compared to the extreme RCPs (2.6 and 8.5) which may incorporate too many uncertainties [60], causing projections to be unreliable. Future scenarios' uncertainty were also accessed through ten Global Circulation Models (GCMs) available in CHELSA; we avoided those with high co-dependency [61], resulting in the selection of MIROC5, CESM1-CAM5, IPSL-CM5A-MR, FIO-ESM, GISS-E2-H, CSIR-O-Mk3-6-0, GISS-E2-R, GFDL-ESM2G, MIROC-ESM-CHEM and MRI-CGCM3 (S1 Table). Each GCM is a model trying to explain how the atmosphere works. We used multiple GCMs to dissolve the effect of one unique GCM and improve predictions [62].

Variables were first submitted to a visual analysis, in which we deleted both the precipitation of the warmest quarter (BIO 18) variable and the precipitation of the coldest quarter (BIO 19) variable due to statistical artifacts, that may not represent the continuous gradient of reality, in the study region. Those artifacts are generated due to a difference in which quarter is the warmest (e.g. BIO18), causing the precipitation of one cell to be the sum from January-February-March, while the very next cell is the summed precipitation from February-March-April. We then masked variables with one degree-wide buffer from each presence record (Fig 1) and excluded variables with a high variance inflation factor (VIF > 3) and highly correlated variables (r > 0.7). This left us with six variables: the mean diurnal range (BIO 2), the isothermality (BIO 3), the mean temperature of the wettest quarter (BIO 8), the precipitation of the wettest month (BIO 13), the precipitation of the driest month (BIO 14) and the precipitation seasonality (BIO 15). These six variables were used to calculate the climatic niche of the species. The selection routine was performed using the usdm package [63] in R 3.6.2 [64]. Models were generated with variables at 30 arc-seconds resolution, while the rasters used to project models were upscaled at a factor of 10, resulting in rasters with a resolution of 2.5 arc-minutes.

## Species distribution modelling

For the SDMs, we applied an ensemble method using the sdm package [65] in R 3.6.2 [64]. We implemented five algorithms using different approaches, with proper pseudo-absence selection, following Barbet-Massin et al. [66], as follows: MaxEnt, a machine-learning approach, with 1,000 randomly selected pseudo-absences; Multivariate Adaptive Regression Splines, a regression-based approach, with 100 randomly selected pseudo-absences; Multiple Discriminant Analysis, a classification approach, with 100 pseudo-absences randomly selected outside a surface-range envelope; Random Forest, a bagging approach, with 224 pseudo-absences randomly selected outside a surface-range envelope; and BIOCLIM, an envelope approach, with 100 randomly selected pseudo-absences. Algorithms were implemented using standard parameterization from the sdm package [65] hence the algorithms were not tuned. Model evaluation was performed with ten runs of a four-fold cross-validation technique (75.00% training and 25.00% test). In each run, we calculated true skill statistics (TSS) and the area under the receiver operating characteristic (AUC). To build ensemble models for each scenario, and after some pre-analysis, we selected models with TSSs and AUCs higher than the mean plus half the standard deviation. The mean AUC value was 0.958, with a standard deviation of 0.059 and a threshold equal to 0.988. The mean TSS value was 0.861, with a standard deviation of 0.112 and a threshold equal to 0.917. Selected models were projected into current and future scenarios and then binarized using the AUC threshold to avoid the use of subjective thresholds. Ensembles from future scenarios were built as a committee average of binarized rasters. Afterwards, we normalized the resulting rasters. This returned an ensemble in which 1 represents sites where all models agree with presences, 0 represents sites where all models agree with absences and the values in between are subject to uncertainty, where 0.5 represents cells with the highest uncertainty (i.e. half of the models agree with an absence, while the other half agree with a presence). We also built three potential refugees for the species by summing the normalized rasters from the five scenarios (current, RCP 4.5/2050, RCP 4.5/2070, RCP 6.0/2050 and RCP 6.0/2070). Then, we applied three thresholds, which were calculated by extracting all values greater than zero from the raster and obtaining the 90th, 95th and 99th quantiles (2.179, 2.850 and 3.930, respectively).

We calculated climatically suitable areas using a weighted method, multiplying the cell's committee average by the cell area and summing all values within the rasters. This conservative method was intended to consider the uncertainty underlying each cell, as well as the different

occupation proportions. Therefore, if a cell had 0.5 value (i.e. 50% chance of the species to occur in the cell), we calculated 50% of the cell area and add it to the total area occupied by the species. We applied this method to all scenarios, as well as, to every country present in the Zambezian and Somalian biogeographical regions [22]. We also masked results from area calculations with the World Database on Protected Areas v. 3.1 polygons [44] to estimate the climatically suitable areas under protection in the regions, countries and scenarios. Area calculations were performed in R 3.6.2 [64].

To get response curves for each variable, we extracted response data from each algorithm and made a regression analysis thorough locally estimated scatterplot smoothing (LOESS), with a span window of 0.5- and one-degree polynomial. This method fits multiple lines using half of the whole data. Each time a line is fitted, we exclude the first record from response data and include the next, until we have included all records.

## Results

Our model demonstrated high performance, with an average AUC of 0.958 (SD = 0.059) and an average TSS of 0.861 (SD = 0.112). Our results show that the probability of *M. tornieri* occurrence increases with an increase of BIO 2, BIO 3, BIO 8, BIO 13 and BIO 15 and started to drop when the optimum condition has reached (S1 Fig). Conversely, the probability of occurrence for the species decreases with an increase in BIO 14 and started to increase after the optimum condition has reached (S1 Fig). However, BIO 3 showed the highest contribution relatively to others (S2 Fig) in predicting the distributional range of pancake tortoise in the Zambezian and Somalian biogeographical regions.

Currently, in the Zambezian and Somalian biogeographical regions, the pancake tortoise has a more extensive range in Tanzania and Kenya than in other countries present in the region (Fig 2). Although there is currently no evidence of records of pancake tortoises in Angola and Ethiopia, surprisingly, the model predicted patches of climatically suitable habitats in those countries under the current climatic scenario (Fig 2). Additionally, the model predicted that the current suitable distribution range of pancake tortoises is 90,668.75 km$^2$, with Kenya contributing 61.10% of the current total range, followed by Tanzania (30.32%), Ethiopia (5.03%) and Angola (3.55%) (Table 1). Considering future climatic scenarios, we predicted that the pancake tortoise's suitable habitat would not decrease. This was observed through the expansion of suitable habitats as predicted by RCP 4.5 and RCP 6.0 (Fig 2 and Table 1). The model predicted that the current distribution range would expand by 303.95% in the year 2050 and 342.47% in the year 2070 for RCP 4.5 and by 278.81% in the year 2050 and 311.99% in the year 2070 under RCP 6.0 (Table 1). Similar to the current scenario, we predict that Kenya and Tanzania will continue to have a larger suitable area (Fig 2 and Table 1) than other countries in the future. However, the distributional range is predicted to expand from the current four countries to ten countries in the future (Table 1).

The highly suitable areas (indicated by higher committee averages) are currently present in Kenya and Tanzania (Fig 2), where the species occurs naturally. In the future, highly suitable habitats will expand into Ethiopia as well (Fig 2); however, the species has not yet been recorded in that country. Although there were observations of pancake tortoises in Zambia (Fig 1), our model predicted that the area is not climatically suitable for pancake tortoises in the current and future scenarios (Fig 2).

Considering protected lands, we found that a larger suitable habitat for pancake tortoises lies outside of the current Protected Areas Network in both current and future climatic scenarios (Table 1). Currently, 32.37% of the suitable pancake tortoise habitat lies inside of protected areas (Table 1). In the future, we predicted that the protected suitable area for pancake

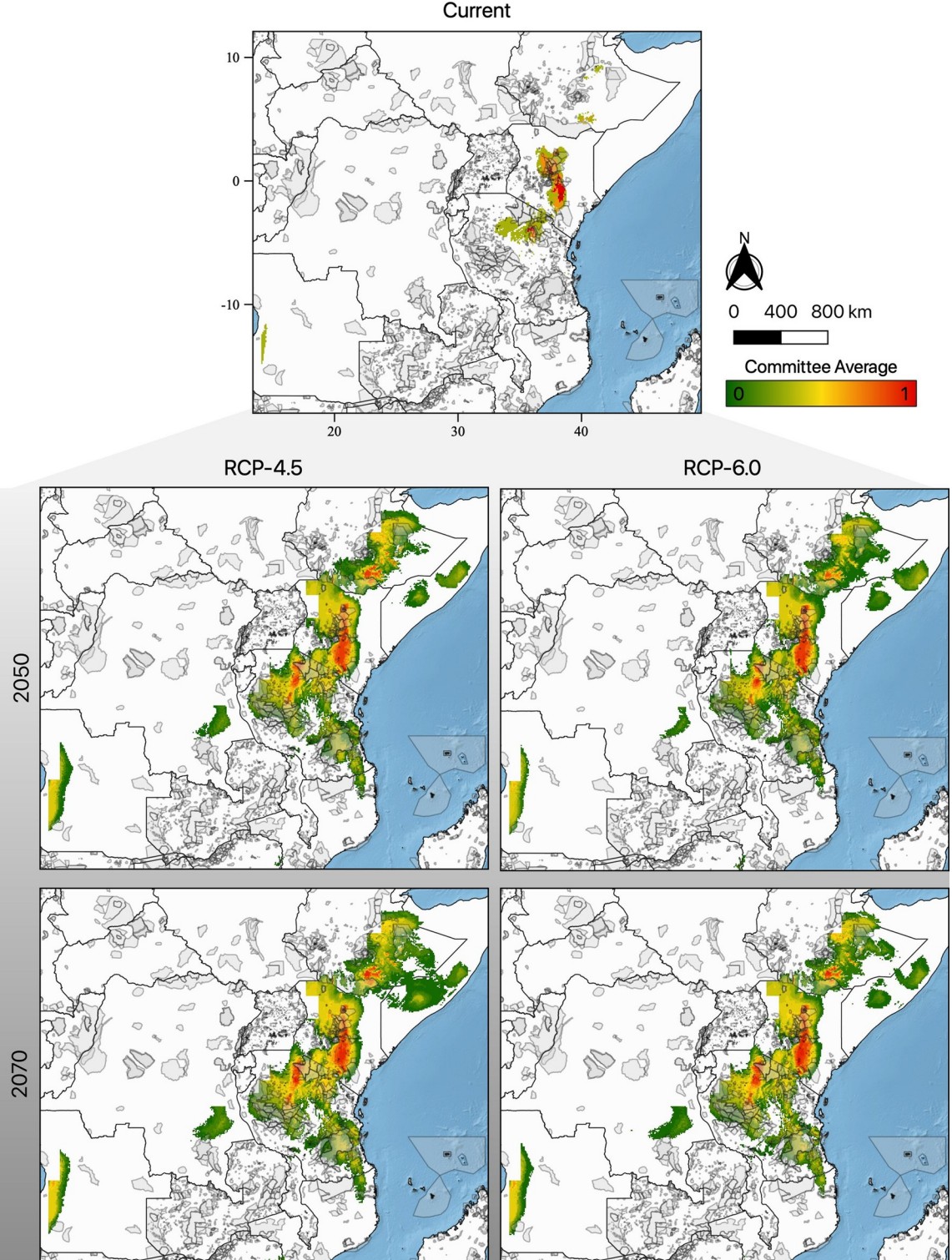

**Fig 2. Distribution of pancake tortoise (*Malacochersus tornieri*) in the Somalia and Zambezian biogeographical regions.** Current and future (2050 and 2070) climatic suitable habitat for pancake tortoise in the Zambezian and Somalia biogeographical regions considering six bioclimatic variables and two future climate scenarios. Warmer colours show more suitable areas, ranging from red to green. Background image was accessed from Natural Earth (public domain): http://www.naturalearthdata.com/.

**Table 1.** *Malacochersus tornieri* **suitable area (km$^2$) in the countries present in the Zambezian and Somalia biogeographical regions for the current and future climate scenarios.**

| Country | Current | | Future | | | | | | | |
|---|---|---|---|---|---|---|---|---|---|---|
| | | | RCP 4.5 | | | | RCP 6.0 | | | |
| | | | 2050 | | 2070 | | 2050 | | 2070 | |
| | Total | % of suitable protected habitat | Total | % of suitable protected habitat | Total | % of suitable protected habitat | Total | % of suitable protected habitat | Total | % of suitable protected habitat |
| Kenya | 55,401.76 | 35.68 | 122,938.04 | 22.19 | 129,259.60 | 21.82 | 121,659.50 | 22.56 | 124,362.90 | 22.48 |
| Tanzania | 27,489.20 | 33.72 | 136,652.13 | 48.95 | 152,800.60 | 48.03 | 126,815 | 46.46 | 149,008.60 | 47.79 |
| Zambia | 0 | 0 | 0 | 0 | 0 | 0 | 0 | 0 | 0 | 0 |
| Mozambique | 0 | 0 | 1,906.56 | 12.75 | 2,424.43 | 16.39 | 996.42 | 6.42 | 1,906.07 | 8.95 |
| Malawi | 0 | 0 | 25.65 | 0 | 34.47 | 0 | 0 | 0 | 38.30 | 0 |
| Zimbabwe | 0 | 0 | 95.31 | 33.34 | 450.11 | 19.67 | 282.20 | 36.24 | 44.29 | 49.99 |
| Angola | 3,219.98 | 0 | 18,528.14 | 1.20 | 19,549.08 | 0.73 | 18,689.62 | 1.50 | 20,321.14 | 1.64 |
| Namibia | 0 | 0 | 0 | 0 | 0 | 0 | 0 | 0 | 0 | 0 |
| Botswana | 0 | 0 | 0 | 0 | 0 | 0 | 0 | 0 | 0 | 0 |
| Somalia | 0 | 0 | 10,233.67 | 0 | 12,181.94 | 0 | 11,891.61 | 0 | 7,692.55 | 0 |
| Ethiopia | 4,557.81 | 6.84 | 71,044.64 | 28.52 | 76,881.24 | 26.89 | 60,338.92 | 29.05 | 65,144.88 | 27.52 |
| Democratic Republic of Congo | 0 | 0 | 4,283.13 | 0.69 | 6,532.77 | 3.30 | 2,327.59 | 4.43 | 4,388.17 | 1.40 |
| Burundi | 0 | 0 | 0 | 0 | 0 | 0 | 0 | 0 | 0 | 0 |
| Rwanda | 0 | 0 | 0 | 0 | 0 | 0 | 0 | 0 | 0 | 0 |
| Uganda | 0 | 0 | 11.15 | 0 | 31.08 | 0 | 44.59 | 0 | 38.85 | 0 |
| South Africa | 0 | 0 | 0 | 0 | 0 | 0 | 0 | 0 | 0 | 0 |
| TOTAL | 90,668.75 | 32.37 | 366,257.22 | 31.39 | 401,179.70 | 30.69 | 343,459.60 | 30.41 | 373,547.30 | 31.50 |
| % of change relative to the current scenario | | | 303.95% | 291.73% | 342.47% | 319.48% | 278.81% | 255.85% | 311.99% | 300.94% |

'Total' is the total suitable area and '% of protected suitable habitat' is the percentage of protected area that overlaps with the species suitable habitat.

tortoises will expand from 114,969.40 km$^2$ (in 2050) to 123,112.71 km$^2$ (2070) in RCP 4.5 and 104,437.30 km$^2$ (in 2050) to 117,672.83 km$^2$ (in 2070) in RCP 6.0 (Table 1), given the current Protected Area Network. However, we predicted that the protected suitable habitat of pancake tortoises will continue to be smaller in the future (RCP 4.5: 31.39% in 2050 and 30.69% in 2070; RCP 6.0: 30.41% in 2050 and 31.50% in 2070; Table 1).

We identified Kenya, Tanzania, Ethiopia and Angola as the countries that maintain the most stable habitat for pancake tortoises over time (Table 2). However, the highest stability occurs within Kenya, Tanzania and Ethiopia (Fig 3), with only Kenya having a highly stable habitat inside the protected areas (Table 2). We predicted that the stable habitats for pancake tortoises within the current Protected Areas Network will continue to be smaller than those of habitats in unprotected areas (percentage of stable habitat present in protected areas: less stable [33.08%], average stability [27.97%] and highly stable [14.87%, present in Kenya only]; Table 2).

## Discussion

We predicted the climatic suitable habitat for pancake tortoises in the Zambezian and Somalian biogeographical regions in the current and future scenarios. The six bioclimatic variables indicated that pancake tortoise occurrence can either increase or decrease until the optimum

**Table 2. Potential climatic stable areas/habitats (in km²) for the pancake tortoise per each country present in the Zambezian and Somalia biogeographical regions.**

| Country | Current suitable habitat (km²) | Less stable | | Mid stable | | Highly stable | |
|---|---|---|---|---|---|---|---|
| | | Total (km²) | % of protected stable habitat | Total (km²) | % of protected stable habitat | Total (km²) | % of protected stable habitat |
| Kenya | 55,401.76 | 92,198.02 | 35.31 | 57,252.66 | 30.89 | 16,066.27 | 15.42 |
| Tanzania | 27,489.20 | 50,153.64 | 35.54 | 16,299.88 | 26.19 | 255.98 | 0 |
| Zambia | 0 | 0 | 0 | 0 | 0 | 0 | 0 |
| Mozambique | 0 | 0 | 0 | 0 | 0 | 0 | 0 |
| Malawi | 0 | 0 | 0 | 0 | 0 | 0 | 0 |
| Zimbabwe | 0 | 0 | 0 | 0 | 0 | 0 | 0 |
| Angola | 3,219.98 | 2,174.18 | 0 | 83.67 | 0 | 0 | 0 |
| Namibia | 0 | 0 | 0 | 0 | 0 | 0 | 0 |
| Botswana | 0 | 0 | 0 | 0 | 0 | 0 | 0 |
| Somalia | 0 | 0 | 0 | 0 | 0 | 0 | 0 |
| Ethiopia | 4,557.81 | 19,904.86 | 20.12 | 8,513.63 | 12.01 | 340.53 | 0 |
| Democratic Republic of Congo | 0 | 0 | 0 | 0 | 0 | 0 | 0 |
| Burundi | 0 | 0 | 0 | 0 | 0 | 0 | 0 |
| Rwanda | 0 | 0 | 0 | 0 | 0 | 0 | 0 |
| Uganda | 0 | 0 | 0 | 0 | 0 | 0 | 0 |
| South Africa | 0 | 0 | 0 | 0 | 0 | 0 | 0 |
| **TOTAL** | **90,668.75** | **164,430.70** | **30.08** | **82,149.83** | **27.97** | **16,662.78** | **14.87** |

Climatic stability was inferred by extracting all values greater than zero from the raster and obtaining the 90th, 95th and 99th quantiles, returning low, mid and high stability, respectively. Area was calculated with a suitability weighted approach. The variation between current suitable habitat and the stable areas is influenced by the weight set in cells where for those of stable areas weight is one, while cells in current suitable area weight is less than one. 'Total' is the total stable area and '% of protected stable habitat' is the percentage of protected area that overlaps with the species stable habitat.

condition has been reached (S1 Fig). This might be due to the fact that reptiles need the optimum condition for laying and hatching their eggs [67]. Therefore, climatic changes can significantly affect reptile's reproductive success [68]. Although isothemality (BIO 3) did have the highest contribution in predicting climatic suitable habitat for pancake tortoise in the Zambezian and Somalian biogeographical regions (S2 Fig), this variable was not selected by Bombi et al. [41] when modelling the distribution of 16 species of Testudinidae in Africa. However, two of our variables (mean temperature of the wettest quarter (BIO 8) and precipitation seasonality (BIO 15)) did match with the one selected by Bombi et al. [41]. These variations could be due to differences in geographical range considered, species involved, number of occurrences and modelling approach.

We predicted that the suitable climatic habitat for pancake tortoises would be less discontinuously scattered in the Zambezian and Somalian biogeographical regions in the future than in current climatic scenarios (Fig 2). The disjointed distribution of pancake tortoises was also observed in the countries in which they currently exist naturally, which are Tanzania [17, 20] and Kenya [18, 20]. We further predicted that the distributional range of pancake tortoises would expand in the future (Fig 2 and Table 1). The expansion of the future distributional ranges of reptiles has also been recorded by Houniet et al. [69] for *Bradypodion occidentale*, González-Fernández et al. [70] for *Thamnophis melanogaster*, Fathinia et al. [71] for *Pseudocerastes urarachnoides* and Sousa-Guedes et al. [72] for 13 different reptile species.

Apart from area expansion, our model also predicted an increase in the number of climatically suitable habitats in countries in which pancake tortoises do not exist naturally from the current two to eight future countries, with Angola being isolated in the far west of the region

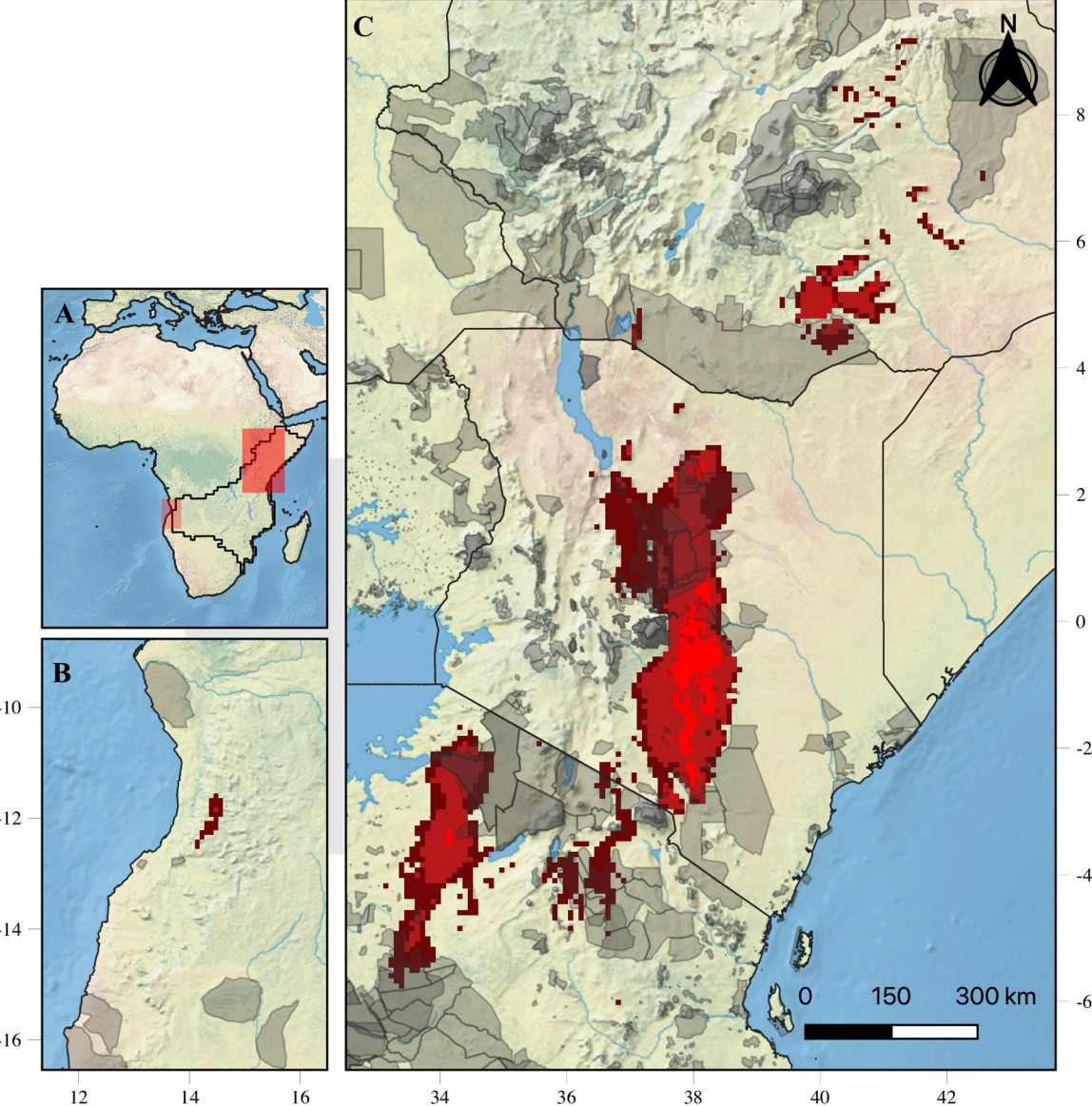

**Fig 3. Potential climatic stable areas for the pancake tortoise in the Zambezian and Somalia biogeographical regions.** (A) Location of stable areas in Africa. (B) Stable areas in Angola. (C) Stable areas in Tanzania, Kenya and Ethiopia. The stable areas were obtained by considering three thresholds from the sum of the five normalized climatic scenarios (current, RCP 4.5/2050, RCP 4.5/2070, RCP 6.0/2050 and RCP 6.0/2070). The brighter red colour indicates the more stable site through time. Background image was accessed from Natural Earth (public domain): http://www.naturalearthdata.com/.

(Fig 2 and Table 1). This scenario of isolation of pancake tortoise populations has been also recorded in Tanzania [50] and Kenya [18], where the species exists naturally. The absence of pancake tortoise in the countries which we predicted to have the climatically suitable habitats could be due to their behaviour of being non-migrant [18–20]. On the other hand, Malonza [18] has suggested that the absence of pancake tortoises in potential habitats is mainly due to elevation, with species occurring from 500–1,800 m above sea level [20]. Pancake tortoises do not occur in some climatically suitable habitats (Fig 2). The prevalence of non-Precambrian rock types between areas where we predict suitable habitat may preclude occupancy by pancake tortoises, as they prefer areas dominated by Precambrian rocks [13].

We did not predict the existence of climatic suitable habitats for pancake tortoise under current and future scenarios in Zambia (Fig 2 and Table 1) although the species have been reported to occur in the country [13]. This could mean that pancake tortoise recorded in Zambia were the result of the international animal trade [19], as a result of which animals from East Africa were exported illegally from the country [20]. However, this argument would require a genetic analysis for confirmation. Conversely, Zambia could be located at the limit of the climatically suitable niche and thus has low climatic suitability, which, when applied at a threshold, turns into an absence.

We predicted that Tanzania, Kenya, Ethiopia and Angola (Fig 3 and Table 1) will have climatically stable habitats over time. As pancake tortoises have not yet been recorded in Ethiopia and Angola, these areas could hold potential for the translocation and introduction of the species. We recommend robust habitat suitability studies of these countries and further quantification of occupancy status given the species apparently occupies suitable habitats in the nearby Zambezian and Somalian biogeographical regions [13, 37].

Protected areas are critical tools for biodiversity conservation [33], yet the African Protected Areas Network offers inconsistent protection to tortoise species [41]. In the Zambezian and Somalian biogeographical regions, only 32.37% of the current climatically suitable area for pancake tortoises fall within protected areas, and this percentage is predicted to decline in the future to 30.41% - 31.50% (Table 1). Additionally, from 66.92% - 85.13% of the stable climatic habitat is predicted to be outside of protected areas. Our results are inconsistent with Bombi et al. [41] argument, who mentioned that the established protected areas in East Africa for wildlife conservation offer sufficient presentation for tortoises. On the other hand, we agree with Bombi et al. [41] findings on pancake tortoise where they found that across the entire range of the species only 22.60% of its range are protected [41]. In Kenya, only 5.00% of the pancake tortoise population is protected, while in Zambia, the species does not occur within the Protected Area Network [20, 21]. Based on occurrence points we collected, in Tanzania, only four out of 22 national parks are occupied by pancake tortoises. The pancake tortoise's suitable habitat is largely unprotected in both the current and the future scenarios, likely increasing the risk of overexploitation and exacerbating negative effects of habitat destruction as in Tanzania [26], Kenya [18, 20] and Zambia [20, 21]. Likewise, ectoparasite prevalence is higher outside of the protected area [73], potentially increasing risk to the species.

## Management implications

Because the current natural range for pancake tortoise does not include some of our current and future predictions for climatically suitable habitats, we recommend future studies be conducted in areas where pancake tortoises do not exist to confirm the absence of the species. As White [45] and Chansa and Wagner [21] have pointed out, pancake tortoises could exist in the entire Zambezian and Somalian biogeographical regions, provided that suitable habitat is present; therefore, confirmatory studies on the existence of the species in the climatically suitable habitats are essential for conservation planning for the species. However, we caution that the existence of pancake tortoises is not solely dependent on the presence of climatically suitable habitats, as Malonza [18] has confirmed the non-presence of pancake tortoises in typical habitats for the species in Kenya. Furthermore, the available suitable and stable habitats outside of the current range could be used as baseline areas for the translocation and introduction of the species where necessary. Therefore, we support the IUCN [27] and Bellis et al. [29], who have suggested the importance of conducting SDMs before translocation and species introduction/re-introduction. Our model did not predict the existence of climatically suitable habitats for pancake tortoises in Zambia (Fig 2). Therefore, we

recommend the maximization of conservation efforts in Zambia in order to maintain the recorded pancake tortoise populations, since they seem to be highly threatened as they are all located outside of protected lands [21].

Furthermore, the presence of a large proportion of the climatically suitable habitat for pancake tortoises outside of protected areas could imply the need for more conservation efforts outside the protected range. These efforts might include the establishment of new protected areas aimed at biodiversity conservation to include suitable habitats for pancake tortoises and therefore minimize anthropogenic impacts on the species [18, 43]. Since the current increase in Protected Area Network have rarely strategically considered global biodiversity maximization [33], establishing protected areas within species suitable habitats could be one of the strategies for protecting global biodiversity. This will also help to reduce the extinction risk for different species under climate change [74].

Area protection, management of international animal trade, species recovery plans and conservation awareness are some conservation actions prioritized by IUCN to save the species under current situation [20]. In addition, majority of the of our predicted suitable habitats and current and future climatic scenarios do fall under highest spatial prioritization for land conservation to minimize extinction risk under climate change in the Afrotropics [74]. Nonetheless, environmental and social context will decide which option is better [75], therefore all conservation stakeholders including local, regional and international organizations, scientists, practitioners and the general public should join forces to save the global biodiversity.

## Conclusion and study limitations

We predict expansion of suitable habitats for pancake tortoises in the future, which may conserve populations of this critically endangered reptile. Importantly, the largest proportion of suitable habitats is outside of the current Protected Area Network, therefore, we suggest the pancake tortoise be upgraded in its listing status from CITES Appendix II to Appendix I.

Because our results were largely based on use of climatic variables, our findings should not be treated as ready-made for on-the-ground application but could be used as one of many tools to help in conservation planning of pancake tortoises. Our decision to use primarily climate variables was because they drive most of the species' distribution [76]. Although Giannini et al. [77], de Araújo [9] and Palacio and Girini [77] have pointed out that the inclusion of biotic factors significantly improves SDMs, we were unable to obtain these data for our study. We recommend that future studies to consider the inclusion of pre-Cambrian rock (as it provides a preferred habitat for pancake tortoises), the international pet trade, land-use changes and ecological interactions as predictor variables. However, in the current situation, it is difficult to obtain these data ready-made for SDM, especially for future scenarios. Furthermore, the application of SDM can be limited by; first, assuming there is a balance between environmental changes and spatial distribution of the species [2, 78, 79], second, the representability of distributional spectrum of the species in relation to the occurrence data used in modelling [78–80], third, data adequacy and resolution applied during modelling, fourth, model performance and fifth, the reliability of climatic future predictions [79, 81].

All in all, our study provides a solid foundation for future development of conservation measures aimed at protecting populations of the critically endangered pancake tortoise.

## Supporting information

**S1 Fig. Response curves from the ensemble models to the six selected bioclimatic variables.** Response curves were fitted through locally estimated scatterplot smoothing (LOESS). Grey

background shows a scale from y-axis which is replicated to every graph.
(DOCX)

**S2 Fig. Variable importance for six less correlated climatic variables of the ensemble species distribution model.** BIO 8 = mean temperature of the wettest quarter, BIO 3 = the isothermality, BIO 2 = mean diurnal range, BIO 15 = precipitation seasonality, BIO 14 = precipitation of the driest month and BIO 13 = precipitation of the wettest month.
(DOCX)

**S1 Table. Ten Global Circulation Models (GCMs) used in our study.**
(DOCX)

# Acknowledgments

We thank Tanzania Wildlife Research Institute (TAWIRI) and Commission for Science and Technology (COSTECH) for granting permit and Tanzania National Park (TANAPA) for giving free access into Tarangire National Park. We also thank Deo Tarimo for his help in obtaining pancake GPS locations from Mkomazi National Park and some from Tarangire National Park. Furthermore, we appreciate all the logistical support which were provided by the College of African Wildlife Management, Mweka during fieldwork. We also thank Nicola van Wilgen and the other anonymous reviewer for their comments which significantly improved this manuscript.

# Author Contributions

**Conceptualization:** Abraham Eustace, Rudolf Mremi, Reginald T. Mwaya.

**Data curation:** Abraham Eustace.

**Formal analysis:** Luíz Fernando Esser.

**Validation:** Patrick K. Malonza.

**Writing – original draft:** Abraham Eustace.

**Writing – review & editing:** Abraham Eustace, Luíz Fernando Esser, Rudolf Mremi, Patrick K. Malonza, Reginald T. Mwaya.

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
