## [Decision Letter · Decision Letter 0]

6 Oct 2020

PONE-D-20-25666

Protected areas network is not adequate to protect a critically endangered East Africa Chelonian: Modelling distribution of pancake tortoise, Malacochersus tornieri under current and future climates

PLOS ONE

Dear Dr. Eustace,

Thank you for submitting your manuscript to PLOS ONE. After careful consideration, we feel that it has merit but does not fully meet PLOS ONE’s publication criteria as it currently stands. Therefore, we invite you to submit a revised version of the manuscript that addresses the points raised during the review process.

Your paper has the potential to make a useful contribution to the literature, but as presented, there are too many gaps for the reader follow your methods through to your conclusions. It is important that you provide enough information about your modeling approach so that the reader can understand how you worked with the species and climate data and that you explicitly describe the outcomes of the critical modeling steps along the way. I have a number of concerns with the modeling that could be addressed during revision, but this will likely require a major overhaul including re-doing the modeling and presenting the information with greater detail and clarity. Both reviewers provide excellent, detailed suggestions to help if you choose to revise. Please note that revision does not guarantee acceptance.

We look forward to receiving your revised manuscript.

Kind regards,

Stephanie S. Romanach, Ph.D.

Academic Editor

PLOS ONE

Journal Requirements:

4. We note that Figures 1-3 in your submission contain map images which may be copyrighted.

a. You may seek permission from the original copyright holder of Figures 1-3 to publish the content specifically under the CC BY 4.0 license. 

Reviewers' comments:

Reviewer's Responses to Questions

**Comments to the Author**

1. Is the manuscript technically sound, and do the data support the conclusions?

Reviewer #1: Partly

Reviewer #2: Yes

2. Has the statistical analysis been performed appropriately and rigorously? 

Reviewer #1: No

Reviewer #2: Yes

3. Have the authors made all data underlying the findings in their manuscript fully available?

Reviewer #1: No

Reviewer #2: Yes

4. Is the manuscript presented in an intelligible fashion and written in standard English?

Reviewer #1: Yes

Reviewer #2: Yes

5. Review Comments to the Author

Reviewer #1: This paper investigates the suitability of future climate space for the critically endangered pancake tortoise, with a view to informing translocations and/or population management that will maximize sustainability of future populations in the face of climate change and other stressors (this species is harvested for the international pet trade). The study finds that similar to the current situation, most future suitable habitat is outside of protected areas. Further they identify additional sites where the species is not currently known to occur that could serve to bolster the population.

While this topic is a very important and the investigation a needed contribution in terms of critically endangered species management, there were some fundamental aspects that relate to species distribution modelling and species conservation that have not been covered/discussed. I have made a number of specific comments, but in general, think that the following need to be addressed to strengthen the paper:

1. More rigorous assessment of the climatic variables selected. How do these relate to the species’ life history and what mechanism of impact are they expected to have on population viability. Also, in terms of the model, what was the relationship with these variables and tortoise occurrence, and what was the relative effect size of each? Are there any potentially informative variables that were not included?

Related to the above, are there any other non-climatic variables for which information exist that might be added? The authors suggest that rock substrate is important. The authors also mention a previously published distribution model. How do the results of these models compare?

2. The tables and figures are useful, but could their utility could definitely be enhanced further with the addition of details. For example, protected areas are not shown and in some instances country borders are also omitted. For the tables, it would be useful to compare current to predicted future situations.

3. The extent to which the current distribution has been impacted by overharvesting is not clear. The current models do not highlight any suitable habitat in Zambia, but only one occurrence was included from the Zambian border. Is it likely that the species used to be more widespread in Zambia? If so, the discussion should rather conclude that lack of current records from the area preclude investigation of habitat suitability in Zambia

4. Some more general discussion/framing of species management under climate change would be useful. There are so many species and large change is expected. Is moving species to far off locations really viable? How are these interventions prioritized? And by whom? The paper by Hannah (2020) details some of the most important corridors for conservation in Africa. There is definite overlap with the areas identified in this study and would be worth mentioning.

Specific comments:

Line 118 – there is no mention (later) of whether your current distribution compares well to this previous model. What the major findings and reasons for any potential differences?

Line 170, in what way would aquaculture threaten the tortoise? I assume this is a grouped categorization by the IUCN, but perhaps leave off the aquaculture part or include in inverted commas?

Lines 179-182 combine sentences to reduce repetition

Fig. 1 Caption: I suggest reordering and rewriting along the following lines to make the buffer area clear from the outset: Current natural occurrence of pancake tortoise (Malacochersus tornieri), based on data obtained in this study, with a 1-degree wide buffer around each presence record, at (A) continental scale and (C) regional scale. (B) shows the Zambezian and Somalia biogeographical region.

Line 233: is some measure of maximum temperature not important from a species perspective under climate change? How close to thermal maxima is the current study area likely to get? How do the variables selected compare to those used in previous studies on tortoises?

242-251 I am not quite sure that I understand the different algorithms – are you saying that you had 5 methods of accounting for pseudo-absences?

Section starting line 241 “Species distribution modelling”– it is not clear at which point the future projections are brought into the modelling. These are referred to as part of a general analysis in line 269, but there is no explicit separation of how current versus future estimates of range were produced.

Line 290 – with regard to the potential habitat in Angola. Was this also found to be the case when pseudo-absences were only generated from within the boundary of the current range? See the description of how different selection of pseudo-absences influences model outcomes and utility in Merow et al. 2013. The selection of pseudoabsences should also be informed by the type of conservation intervention that is most feasible. For example, I would imagine that conservation interventions within the species current countries of occupancy would be more likely to succeed than attempts in countries far beyond where the species has ever occurred. Should the area in Angola really be considered as part of the conservation strategy? The species has no means to natural dispersal to reach the area in Angola, so any such intervention would need to be human mediated, and require significant international collaboration. Is this feasible? If not, I would concentrate statistics on countries in the proximity of the species current range and leave additional areas as a passing mention (unless the politics of such species/climate change management is dealt with in more depth).

Having said this, what happens at the border of the Zambezian Somalia bioregion? Is the habitat entirely unsuitable? What do future models beyond this border suggest?

Table 1 – please indicate whether the area protected is the protected area that overlaps with the species distribution/potential distribution or whether it indicates the total area under protection in a particular country. I assume the former. It may also be useful to provide the protected area as a percentage. Combining the information in Tables 1 and 2 to indicate the proportion of PA that remains stable might also be useful?

Lines 319-333 and Tables. It is hard how to determine how the much larger numbers and percentages presented in lines 319-325 compare to the smaller percentages of stable protected area in lines 326-333. It might be better to concentrate the results on the area within protected areas currently suitable for these tortoises and how stable this remains over time, as well as loss and addition of potential new protected area that might become suitable?

Table 2: It would also be useful to add information on the current situation to Table 2 for comparison purposes (unless these numbers are intended to sum to the current total)? Is there any protected area that was suitable that becomes entirely unsuitable?

Line 312, Fig. 2 – it is hard to tell how suitable Zambia is without the inclusion of country borders on the map. Is the model working correctly if it does not predict currently suitable habitat in areas where observations have been made? What is so different about Zambian habitat that causes it not to be selected? Plotting part A and B of Figure 1 together might assist in identifying where the edge of the bioregion is in relation to the country borders.

Results: There are some key factors missing from the current results/discussion section:

• Why are the core drivers of the model not discussed? It would be useful to see response curves for individual variables, as well as effect sizes

• Related to the above, the discussion should include some mechanistic drivers of the potential patterns – why are the key variables the ones picked up by the analysis

• Are there any potential climate variables thought to influence the species breeding or foraging success that have not been included in the current models?

• Are soil data not available? It would be useful to include these data in the models if they are available.

• How does the representation of the current distribution compare to other models done for this species in the past

• How stable does the area currently under protection remain? To what extent is climate change really expected to add to the threat of the species? It appears that conservation of the species outside of the protected area is currently a priority regardless of climate change, but the extent to which climate change may change this in future is what needs to be discussed. Is there anywhere in particular, considering land-use and potential for other threats, where you would recommend the establishment of new protected areas to protect this species? [This would be in areas least impacted by climate change and most suitable for the species].

For the reported suitable ranges (lines 290-300), how was ‘suitable area’ quantified? I understand from the methods, there was a committee averaging technique applied, but this resulted in cases where some of the colours represent almost no agreement of presence. Which degree of consensus of suitable area (or colour on the map) was classified as ‘suitable’ for the numbers reported in the Tables?

Fig. 2 There does not appear to be a large difference between the RCPs in terms of high match areas (>0.5; yellow to red). Why might this be?

Fig. 3 please indicate the location of protected areas on this map

Merow, C., Smith, M.J. & Silander, J.A. (2013) A practical guide to MaxEnt for modeling species’ distributions: what it does, and why inputs and settings matter. Ecography, 36, 1058-1069.

Hannah, L., Roehrdanz, P.R., Marquet, P.A., Enquist, B.J., Midgley, G., Foden, W., . . . Svenning, J.C. (2020) 30% land conservation and climate action reduces tropical extinction risk by more than 50%. Ecography.

Reviewer #2: General comments:

This paper examines the distribution of critically endangered pancake tortoises. The authors used occurrences obtained from Tanzania, Kenya and Zambia to estimate suitable and stable areas in other countries (projecting/predicting). They also used a protected area network to assess how many of the suitable and stable areas are protected for the conservation. The authors conclude most of the identified suitable habitats remain outside of protected areas. I think this is a sound contribution to the literature. I have provided specific comments below, aimed at clarification of methods and results. In addition, particularly in the discussion, I have suggested ways the text can be streamlined to flow better and be more concise and direct. Finally, I think some added text addressing some of the problematic areas of SDM at the end of the discussion might make this paper stronger. Specific comments are below, but in general I am referring to how SDMs are indeed frequently used, but they are also very controversial. So I just suggest you consider briefly addressing this and indicate how your study deals with these issues (e.g., by careful selection of data points etc.).

Specific comments:

Line 43: It seems like a few more citations might be helpful to demonstrate the growing interest in species distribution modeling.

Line 53: Additional citations seem warranted here.

Line 54: Provide citation, if possible.

Line 81: Consider starting the sentence differently because the previous sentence also beings with “although”.

Line 86: It would be more concise to say “…has indicated climate-matching of recipient sites is important for understanding suitability of these areas for introduced/translocated species”, or something similar. In general, this sentence is a bit cumbersome and seems to be discussing translocation, so you might want to set that up before this sentence with a better transition. Maybe just lead in with something simple transitioning from “no study has assessed climate change in these tortoises” to why re-introductions might be necessary.

Line 93: I think this phrasing is a little problematic “because SDM is the most widely accepted method of predicting climatically suitable habitats”. I might eliminate this part. SDM is frequently used, true, but it is also very controversial, and refers to many different methods, including occupancy modeling. So SDM ends up being an umbrella term for several of different ways of mapping species distribution.

Line 97: What is the implication for “there are endangered species that inhabit areas outside of protected lands”? I think this information will help make a stronger point. In essence—what is the point you are trying to make? I would suggest moving Lines 102-104 to the beginning of the paragraph, so the main point is clearer from the beginning. Only slight sentence re-structuring would be needed for the rest of the paragraph if you re-arranged this way.

Line 112: See earlier comment and consider removing “SDMs are essential”. This part is unnecessary. What is essential is the examination of how well protected areas perform, not the SDM. Plenty of other methods can be used to achieve the same goal.

Lines 113-114: Correct, and there is a lot of controversy over SDM’s, for example, in the data used to generate them (see below). I am not saying you need to reference these papers, I’m just illustrating a point that drastically different results occur with even small differences in data inputs. I wonder if controversy should be addressed here, or perhaps in the discussion in a way that shows you have thought about this and carefully selected your data and approach.

• Rodda, G. H., C. S. Jarnevich, and R. N. Reed. 2009. What parts of the US mainland are climatically suitable for invasive alien pythons spreading from Everglades National Park? Biological Invasions 11:241–252.

• Pyron, R. A., F. T. Burbrink, and T. J. Guiher. 2008. Claims of Potential Expansion throughout the U.S. by Invasive Python Species Are Contradicted by Ecological Niche Models. PLoS ONE 3:e2931

• Rodda, G. H., C. S. Jarnevich, and R. N. Reed. 2011. Challenges in Identifying Sites Climatically Matched to the Native Ranges of Animal Invaders. PLoS ONE 6:e14670.

Line 125: “the occurrence of more stable areas over time” seems like it needs to be more explicitly defined, but hopefully this will be more clear in the methods.

Line 127: Consider re-phrasing as “This study may inform specie management approaches, including identifying suitable areas for translocation and the establishment of nature reserves where species can be protected with minimal human intervention [10]. This way is more concise (the sentence is long). But more importantly, the way you have this written reads as a promotion for SDM’s and I don’t think that is warranted without more information on how they need to be used carefully. The discussion probably needs this information.

Line 159: Remove “up”

Lines 160-162: “…tortoises are disjointly distributed from the northern to southern areas, passing though the central to south-eastern regions of the country”. This reads awkwardly and is a bit confusing.

Line 175: Remove “on the ground”

Lines 175-178: Can you reference one of your maps for where your field sites were, generally? Or is this best left unsaid because of poaching concerns? I mention this because many will not know where these areas are without a visual.

Lines 185-187: I like that you specified how you cleaned the records and would say that as much information here as possible is warranted, because even one incorrect data point can throw off model results.

Lines 188-190: It is not clear why you would need elevation data to get coordinates. Please clarify. Further, is this a source of error? These are extrapolated data and could inflate errors in your predictions. Again, I think some careful explanation of how these models are sensitive to user inputs, and how you addressed this carefully in your approach is important.

Lines 216-219: Provide a citation indicating they are more realistic.

Line 214: What is a Representative concentration pathway?

Line 219: Variations in future scenarios? Please consider adding a little more detail.

Line 220: Are you saying you ran 10 models to quantify variation? Please clarify.

Lines 222-223: What are all these acronyms? Define them as models and list out what they stand for or provide a citation that identifies them.

Line 225: What is BIO 18? Bio 19? Clarify.

Lines 224-230 are confusing. What are you doing? Perhaps say something like “To evaluate different models we first eliminated x, because of …., and then eliminated y, to assess …..” Or some similar phrasing.

Line 252: What standard procedures?

Line 261: How does this avoid subjective thresholds? Perhaps re-phrase as “The selected models were binarized using the AUC threshold to avoid use of a subjective threshold”. Include citation?

Line 272: Consider removing “furthermore”.

Line 284: You are letting R average the models built with different algorithms, which is fine. I just wonder if you can extract the information and depict a table of each algorithm’s relative performance. This is just a suggestion.

Line 300-301: You say “will continue to” and “will expand” etc. Clarify that you are predicting these things with “we predict that…” for example. These results are just predictions, not an indication of what will happen with 100% certainty. Fix this throughout the Results where applicable. E.g., Lines 320, 323, 331

Line 329: Define stability in this context; likewise define in Table caption on Line 341. How did you determine stability?

Line 335: Is ‘red sheds’ a typo?

Line 345: Begin with “We predicted that…” rather than “Our SDM predicted”

Line 359: Reads awkwardly. What do you mean by “the isolation of pancake tortoise populations is also occurring within Tanzania…”? Does this refer to future predictions? Should you re-phrase as “Based on our model predictions we expect tortoise populations to become isolated…”?

Line 361: Re-phrase, reads awkwardly.

Line 363: “Furthermore” is not necessary and reads a little awkwardly in this context.

Lines 356-370 could use some tightening up; the flow if the text is a bit choppy and could be streamlined a bit. Instead of “As pancake tortoises prefer areas featuring Precambrian rocks, the presence of other rock types between the suitable habitats could act as a distribution barrier [13]. Therefore, this could be another reason for the non-existence of pancake tortoises in some climatically suitable habitats; however, the species may occur in the Zambezian floral region, provided that suitable habitat is available” try “Pancake tortoises do not occur in some climatically suitable habitats (citation/Fig?). The prevalence of non-Precambrian rock types between areas where we predict suitable habitat may preclude occupancy by Pancake tortoises, as they prefer areas dominated by Precambrian rocks [13].”

Line 370: This is confusing “however, the species may occur in the Zambezian floral region, provided that suitable habitat is available [13,37]” given the previous statement saying that they often don’t occur in suitable habitat.

Lines 371-383: As with the previous comment this text could be re-written to be more concise and to flow better.

Line 387: Remove “Additionally”. Also, as above, this sentence could use some restructuring for conciseness and clarity. For example, I might re-phrase as follows: “We recommend robust habitat suitability studies in these countries and further quantification of occupancy status given the species apparently occupies suitable habitats in the nearby Zambezian and Somalian biogeographical regions [37],[13].”

Line 391-393: Consider re-phrasing as “Protected areas are critical tools for biodiversity conservation [23], yet the African Protected Areas Network offers inconsistent protection to tortoise species [33].

Line 402: “none of the recorded species is within” reads awkwardly. Consider re-phrasing as “while in Zambia, the species does not occur within the Protected Area Network [12,13].”

Line 403: Consider re-phrasing as “In Tanzania, only 4 out of 22 national parks are occupied by pancake tortoises.” Also, is this based on your results? If so, clarify with something like “we predict that in Tanzania only 4 of 22 parks….”

Lines 404-409: Consider re-phrasing for better flow: “The pancake tortoise’s suitable habitat is largely unprotected in both the current and the future scenarios, likely increasing the risk of over exploitation and exacerbating negative effects of habitat destruction as in Tanzania [18], Kenya [10,12] and Zambia [12,13]. Likewise, ectoparasite prevalence is higher outside of protected area [59], potentially increasing risk to the species”.

Line 412: Consider re-phrasing as “Because the range of Pancake tortoises does not include current and future predictions for climatically suitable habitats, we recommend future studies be conducted in areas where pancake tortoises do not exist to confirm absence of the species.”

Line 444: “or even their inversion into a trend of growth.” This part seems a bit speculative at this point. I would consider removing, but if you keep, perhaps consider re-phrasing this sentence with something similar to: “We predict expansion of suitable habitat for pancake tortoises in the future, which may conserve populations of this critically endangered reptile”.

Line 444-448: This is an important sentence. I would re-phrase as “Importantly, the largest proportion of suitable habitat is outside of the current Protected Area Network, therefore we suggest the Pancake tortoise be upgraded in its listing status from CITES Appendix II to Appendix I.” This is more streamlined and direct.

Line 449-452: Re-phrase as “Because our results were largely based on use of climactic variables, our findings should not be treated as ready-made for on-the-ground application but could be used as one of many tools to help in conservation planning of Pancake tortoises.” This is a bit more streamlined and concise.

Line 453: Re-phrase as “Our decision to use primarily climate variables was because climate variables drive most of the specie’s distribution”. This is a lot more concise, or less repetitive.

Line 462: I think a stronger ending is warranted. I might suggest saying “Our study provides a solid foundation for future development of conservation measures aimed at protecting populations of the critically endangered pancake tortoise.” or something similar.

6. PLOS authors have the option to publish the peer review history of their article (what does this mean?). If published, this will include your full peer review and any attached files.

Reviewer #1: **Yes: **Nicola van Wilgen

Reviewer #2: No

---

## [Author Response · Author response to Decision Letter 0]

24 Dec 2020

Please, see the attached document named 'PLOS ONE Reviewer comments and responses' for response to editor and reviewers comments.

---

## [Editor Report · Decision Letter 1]

7 Jan 2021

Protected areas network is not adequate to protect a critically endangered East Africa Chelonian: Modelling distribution of pancake tortoise, Malacochersus tornieri under current and future climates

PONE-D-20-25666R1

Dear Dr. Eustace,

We’re pleased to inform you that your manuscript has been judged scientifically suitable for publication and will be formally accepted for publication once it meets all outstanding technical requirements.

Kind regards,

Stephanie S. Romanach, Ph.D.

Academic Editor

PLOS ONE

Additional Editor Comments (optional):

- e.g., L 379, L 455: "Pancake" is capitalized but elsewhere is lowercase. Please check manuscript text for consistency.

- Fig S1 middle left panel x-axis label is incomplete. Perhaps remove "of" to fit all text or expand figure size or reduce font size.

- Table S1: should read "GCMs" not GCM's"
---

## [Editor Report · Acceptance letter]

11 Jan 2021

PONE-D-20-25666R1 

Protected areas network is not adequate to protect a critically endangered East Africa Chelonian: Modelling distribution of pancake tortoise, *Malacochersus tornieri* under current and future climates 

Dear Dr. Eustace:

I'm pleased to inform you that your manuscript has been deemed suitable for publication in PLOS ONE. Congratulations! Your manuscript is now with our production department. 

Kind regards, 

on behalf of

Dr. Stephanie S. Romanach 

Academic Editor

PLOS ONE